# Significance of Arbuscular Mycorrhizal Fungi in Mitigating Abiotic Environmental Stress in Medicinal and Aromatic Plants: A Review

**DOI:** 10.3390/foods11172591

**Published:** 2022-08-26

**Authors:** Abir Israel, Julien Langrand, Joël Fontaine, Anissa Lounès-Hadj Sahraoui

**Affiliations:** Unité de Chimie Environnementale et Interactions sur le Vivant (UCEIV-UR 4492), Université Littoral Côte d’Opale, SFR Condorcet FR CNRS 3417, CS 80699, F-62228 Calais, France

**Keywords:** medicinal and aromatic plants, essential oil, abiotic stresses, arbuscular mycorrhizal fungi, amendments

## Abstract

Medicinal and aromatic plants (MAPs) have been used worldwide for thousands of years and play a critical role in traditional medicines, cosmetics, and food industries. In recent years, the cultivation of MAPs has become of great interest worldwide due to the increased demand for natural products, in particular essential oils (EOs). Climate change has exacerbated the effects of abiotic stresses on the growth, productivity, and quality of MAPs. Hence, there is a need for eco-friendly agricultural strategies to enhance plant growth and productivity. Among the adaptive strategies used by MAPs to cope with the adverse effects of abiotic stresses including water stress, salinity, pollution, etc., their association with beneficial microorganisms such as arbuscular mycorrhizal fungi (AMF) can improve MAPs’ tolerance to these stresses. The current review (1) summarizes the effect of major abiotic stresses on MAPs’ growth and yield, and the composition of EOs distilled from MAP species; (2) reports the mechanisms through which AMF root colonization can trigger the response of MAPs to abiotic stresses at morphological, physiological, and molecular levels; (3) discusses the contribution and synergistic effects of AMF and other amendments (e.g., plant growth-promoting bacteria, organic or inorganic amendments) on MAPs’ growth and yield, and the composition of distilled EOs in stressed environments. In conclusion, several perspectives are suggested to promote future investigations.

## 1. Introduction

Medicinal and aromatic plants (MAPs) are a diverse group of plant species that have received a great deal of interest due to their herbal medicine, pharmaceutical, cosmetic, and nutritional applications [1,2,3]. While medicinal plants include herbs used for therapeutic or pharmacological purposes, aromatic plants are rich in aromatic compounds and provide products that are widely used as spices or flavoring agents, and in cosmetics, or medicine [4,5]. These plant species are mainly cultivated for their various parts (roots, stems, leaves, flowers, seeds, buds, rhizomes, wood, bark, etc.) containing a wide range of secondary metabolites, including essential oils (EOs), which exert antimicrobial, antioxidant, and anti-inflammatory effects [6,7,8]. The cultivation of MAPs is looked upon not only as a source of effective health care products but also of income and livelihood security [9]. The World Health Organization estimates that 80% of the world’s population depends on medicinal herbs for primary healthcare and wellness [10,11]. The use of MAPs has increased widely over the past three decades, and partly because of their low side effects, their demand has expanded not only in local but also international markets [12,13,14].

With increasing global warming, the intensity of droughts and heatwaves will increase, as along with other abiotic stress conditions such as salinity and flooding [15,16]. These various environmental stresses can severely affect the growth, development, and productivity of MAPs [17]. Indeed, MAPs are not tolerant of the impacts of climate change [18]. It has been reported that various abiotic stresses influence the yields and the chemical composition of EOs and lead to variations in their quality [19,20,21]. The chemotype of EOs extracted from cumin (*Cuminum cyminum* L.) seeds changed under water stress, resulting in a change in the EO’s odor [22]. Furthermore, a reduction was observed in the relative concentration of 1,3,8-*p*-menthatriene in the EO extracted from plain-leaved parsley, *Petroselinum crispum*, exposed to water stress [20]. This decrease could be detrimental to oil quality.

Due to the significance of MAPs for humans and the detrimental impacts of abiotic stress on MAP’s growth and productivity, to improve plant growth and productivity there is a need to identify adequate approaches to enhance plant tolerance to abiotic stress. Amongst eco-friendly agricultural practices, the addition of microbial and/or organic or inorganic amendments could be sustainable mitigation strategies to fight against the detrimental effects of abiotic stress [23,24,25,26,27,28].

Plants rely greatly on root-associated microorganisms, particularly bacteria and fungi, to improve plant performance under abiotic stress [29,30]. Arbuscular mycorrhizal fungi (AMF) of the phylum Glomeromycota are known to establish a symbiotic relationship with the roots of over 80% of terrestrial plant species [31], including MAPs [32,33,34]. Such symbiosis helps plants to obtain from the soil more water, and mineral nutrients including phosphorus, nitrogen, and oligo-elements. In return, the AMF takes photo-assimilates (i.e., carbohydrates) from the host plant [31,35,36]. It has been established through many studies that the inoculation of MAPs by AMF renders them more tolerant to abiotic stress conditions including water stress, salinity, and pollution [32,33,37,38], through the involvement of several mechanisms at morphological, physiological, and molecular levels [38,39,40,41,42,43,44,45].

The synergistic effect of microbial (i.e., AMF, plant growth-promoting bacteria (PGPR)) and organic or inorganic (i.e., compost, biochar, manures, NPK, silicate, and clay) soil amendments is beneficial for plant tolerance, by enhancing physicochemical properties and total organic matter content, as well as by increasing nutrient availability and the water-holding capacity of the soil [46,47,48]. Under abiotic stress such as water stress, salinity, or pollution, with the presence of microbial or organic or inorganic soil amendments, the improvement of plant growth and secondary metabolite production has been reported for various MAPs such as pennyroyal (*Mentha pulegium*), sage (*Salvia officinalis)*, sweet basil (*Ocimum basilicum*), rosemary (*Rosmarinus officinalis)*, and cumin (*Cuminum cyminum*) [49,50,51,52]

The present review aims first to analyze the data of several studies reporting the impacts of various abiotic stresses (water stress, salt stress, low and high temperatures, light, and pollution) on MAPs’ growth, and on the yield and composition of distilled EOs. Secondly, it deciphers the AMF-mediated mechanisms to improve MAPs’ tolerance at morphological, physiological, and molecular levels. Finally, the role of PGPR with organic and inorganic amendments in alleviating abiotic stress is discussed. In conclusion, several perspectives to promote future investigations are suggested.

## 2. Impact of Abiotic Stresses on MAPs

### 2.1. Water Stress

Water stress is among the most important abiotic factors that severely affect MAPs’ growth, and it can adversely affect yield and EO composition [20]. Various studies have established that water stress causes a reduction in the vegetative growth of MAPs (Table 1). For instance, in lavender species such as *Lavandula angustifolia* [19] and *Lavandula stoechas* [53], water stress reduced plant height, leaf area, and number of leaves per plant. In addition, Alishah et al. [54] showed that water stress decreased the plant height, stem diameter, number, and area of leaves for sweet basil (*Ocimum basilicum*). The restriction in shoot growth was most likely due to carbohydrate reallocation in favor of root growth, or to a decrease in photosynthesis efficiency [55]. The water stress effect can be extended to the reproductive stage by decreasing seed yield.

Numerous studies have reported that water stress has a negative effect on the number of umbels and umbellets during the reproductive stage in various Apiaceae species, such as cumin (*Cuminum cyminum* L.) [22], sesame (*Sesamum indicum* L.) [64], caraway (*Carum carvi* L.) [56], and coriander (*Coriandrum sativum*) [57]. Concomitant with this, reductions in seed yield were observed.

Several studies have shown that water stress modifies EOs yield and composition by increasing or decreasing constituent levels depending on plant species and stress duration. Water stress has been found to have a positive effect on the yield of EOs distilled from MAPs including Iranian native savory (*Satureja hortensis*) [63], Mexican oregano (*Lippia berlandieri Schauer*) [65], basil (*Ocimum* sp.), and caraway (*Carum carvi* L.) [56] (Table 1). It has been reported that the EOs of Greek sage (*Salvia fruticose)* doubled under moderate water stress and increased by 83% under severe water stress, compared with normal conditions [19]. In another sage species, *Salvia officinalis*, Bettaieb et al. [62] reported an increase in EO yield when plants were cultivated under water stress. The effects of water deficit on the EOs of three parsley (*Petroselinum crispum*) cultivars, designated as plain-leafed, curly-leafed, and turnip-rooted, were studied by Petropoulos et al. [20]. They showed an enhancement in the EO yield for plain-leafed and curly-leafed parsley but not for turnip-rooted parsley. There is evidence that the presence of a high density of oil glands due to limited leaf area might explain such a rise in EO content [66]. Conversely, a decrease in EO yield was reported in a number of MAPs, including rosemary (*Rosmarinus officinalis* L.) [53,60], anise (*Pimpinella anisum* L.) [21], lavender (*Lavandula latifolia*) [19], and clary sage (*Salvia sclarea*) [61].

From the aforementioned studies, it is clear that water stress also influences the EO composition. Water stress was found to induce changes in the composition of EOs distilled from cumin (*Cuminum cyminum* L.) [22]. Furthermore, water stress modified the chemotype of the EO from γ-terpinene/phenyl-1,2 ethanediol in control seeds to γ-terpinene/cuminaldehyde in stressed seeds, possibly changing the EO’s odor [22]. However, the EO chemotype of caraway (*Carum carvi* L.) did not change under water stress, but the content of limonene and carvone increased in the oil extracted from caraway seeds [56]. Water stress decreased certain compositions within the EO extracted from plain-leafed parsley, particularly 1,3,8-*p*-menthatriene [20]. As a result, this decrease could be detrimental to oil quality [67], although an increase in myristicin, which is another essential aromatic constituent, was observed [20]. Under water stress, a decrease in total monoterpenes and an increase in oxygenated monoterpenes were observed in lavender (*Lavandula angustifolia*) and Greek sage (*Salvia fruticose*) [19]. This variation in EO yield and composition might be due to enzyme activity and metabolism enhancements [22,68,69].

### 2.2. Salt Stress

Salt stress, generally resulting from poor quality irrigated water, is considered a major environmental factor limiting plant growth and productivity [70]. There are several common features between salt and water stress, as in both cases, the primary effect is a lower water potential in the soil around the roots [71].

Many researchers have studied the effect of soil and water salinity on the growth of MAPs, as well as on the yields and composition of their distilled EOs. A decrease in shoot and root dry weight, as well as seed yield in response to salinity, was reported in sweet fennel (*Foeniculum vulgare*) [72] and ajwain (*Trachyspermum ammi* L.) [73]. Many other plant growth parameters including plant height, root length, number of leaves per plant, branches per plant, and flowers per plant decreased in a number of MAPs under salt stress (Table 2).

From a wide array of experiments, it has been demonstrated that EO yield and composition can vary with plant species, cultivation conditions, and salt concentrations in the soil. An increase in EO yield was observed for English marigold (*C**alendula officinalis*) [78], thyme (*Thymus vulgaris*) [88], and sweet basil (*Ocimum basilicum* L.) [90]. The increase in EO yield was attributed to a higher oil gland density [66]. In contrast, other studies have revealed that high salinization of soil and water had a negative effect on oil yield for several MAPs including sage (*Saliva officinalis*) [84], chamomile (*Matricaria chamomile*) [77], mint (*Mentha* sp.) [80], and marjoram (*Origanum majorana*) [82] (Table 2). In sage (*Saliva officinalis*) leaves and fruits, the EO yield increased under moderate salt stress (50 and 75 mM), but when the NaCl concentration was increased to 100 mM, the oil yield decreased [83,84]. The application of salt stress induced marked changes in the major EO compounds of sage (*Saliva officinalis*) fruits [84], in particular viridiflorol, which acts as an antifungal potential, and manool, which is a precursor for aromatic products and has an ambergris odor [91]. In control plants and at a low concentration of NaCl (25 mM), viridiflool was the main compound in EOs, whereas at a high salt concentration (100 mM) manool was the dominating compound. These variations might be due to the induction of enzyme activities involved in the biosynthesis of the latter compounds [84]. In coriander (*Coriandrum sativum* L.) leaves, the EO yield was increased at 25 and 50 mM NaCl concentrations while it decreased at a high NaCl concentration (75 mM) [74]. On the other hand, in coriander fruits and roots, the EO yield increased with increasing NaCl concentration (0, 25, and 50 mM) [75,76]. The compositions of EOs from coriander leaves and roots were affected differently by salt treatment. Coriander fruit contains linalool and camphor as major constituents, and their amounts increased along with the NaCl concentrations.

### 2.3. Low and High Temperatures

Temperature is one of the most important abiotic factors limiting plant growth and productivity [92]. Heat stress (high temperature) and cold stress (low temperature) can cause a series of changes at physiological, biochemical, and molecular levels [93]. Heat stress induces leaf senescence, cell membrane damage, degradation of chlorophyll content, and denaturation of various proteins [94]. Several studies are available on the effect of temperature stress on MAPs’ growth [95,96] as well as on EOs’ quantity and quality [97,98]. The effect of high temperature on sweet basil (*Ocimum basilicum*) was studied by Al-Huqail et al. [95]. The authors observed inhibition in plant growth under all high temperature treatments (35, 45, and 55 °C) compared with the control treatment (25 °C), which might be due to the effects of high temperature on plant metabolism [95]. Similarly, plant growth parameters (flower yield and plant height) for chamomile (*Matricaria chamomilla*) decreased at high temperatures (15, 20, and 25 °C) while these parameters increased at a low temperature (12 °C) [97]. In contrast, the fresh and dry weight of fennel (*Foeniculum vulgare*) were reduced under cold stress (2 °C) [96]. Another study has shown that increased temperature can cause early onset of senescence in American ginseng (*Panax quinquefolius*) and, subsequently, decreased photosynthetic rate. Furthermore, a decline in total biomass was observed [99]. The decrease in plant growth parameters might be because stomata become closed or partially closed under temperature stress, which causes a reduction in the CO_2_ availability, and consequently the efficiency of photosynthesis decreases [97,100]. In addition, the reduction in leaf area and stomatal conductance affects the plant’s capacity to intercept light and capture CO_2_ and, consequently, its ability to produce biomass [97,101].

Temperature stress can also affect the production of secondary metabolites including EOs [93,97]. For example, at low temperatures the EO yield of chamomile (*Matricaria chamomilla*) increased, while it decreased at a high temperature [97]. In another study, the EO yield of chamomile (*Matricaria chamomilla*) increased at 10 and 20 °C and decreased under cold stress (5 °C) [102]. Some studies also showed that EO composition can be affected by low temperature. For example, Nguyen et al. [98] indicated an increase in some compounds such as eugenol, methyl eugenol, and β-caryophyllen in the EO of holy basil (*Ocimum tenuiflorum*), cultivated at low temperatures. Similarly, Rastogi et al. [103] reported that eugenol and methyl eugenol content in holy basil (*Ocimum tenuiflorum*) EOs decreased under cold stress.

### 2.4. Light

As well as being the primary source of energy for photosynthesis, light also acts as a signal and can regulate plant development [104,105]. Different parameters such as light quality (wavelengths), quantity (fluence rate), and duration strongly influence plant growth and development, as well as plant productivity [106,107,108]. Although light provides energy for photosynthesis and regulates plant development, it may also function as a stress factor [108]. Many experiments have been conducted to assess how light quality, quantity, and duration affect the growth and EO yield of MAPs. For instance, the fresh weight of mint (*Mentha* spp.) was highest under combined red and blue LED light (70/30%) compared to normal growing conditions [109]. Likewise, the combination of red and blue LED light increased the fresh and dry weights of the shoots and the leaf numbers of lemon balm (*Melissa officinalis*) compared with red light alone, blue light alone, or white light [110]. Another study showed that the leaf and shoot numbers of coriander (*Coriandrum sativum*), as well as its fresh and dry mass, increased under different ratios of red: blue compared with red light alone [111]. In contrast, the total fresh lateral-shoot weight of sweet basil (*Ocimum basilicum*) was highest under blue light compared with other lights (red, green, blue-green, or white light) [112]. The combination of blue and white light increased the leaf area of peppermint (*Mentha piperia*) [113]. These studies demonstrate that combined red and blue light is more effective than monochromatic red light for plant growth, and their conclusions explain that the red light wavelength (650–665 nm) matches the absorption peak of chlorophyll a/b found in the chloroplast [114], while blue light has a complementary effect [115]. Furthermore, blue light has been shown to affect the opening of stomata [116], allowing more CO_2_ entrance for photosynthesis, which is reflected by an increase in dry matter [117]. The effect of increasing light intensity (approximately 4, 7, 11, or 20 mol m^−2^ d^−1^) on the growth of African basil (*Ocimum gratissimum*) was conducted by Fernandes et al. [118]. Results from that experiment indicated that the number of leaves, leaf area, and plant height increased up to 10 mol m^−2^ d^−1^, but decreased after this value. In the presence of green LED light, the fresh weight and leaf area of sweet basil (*Ocimum basilicum* L.) were higher than under other tested lights [112].

Ultraviolet (UV) light is stressful for plants, causing dwarfing and loss in photosynthesis, but appropriately using UV light can increase plant yield in some species [119]. Supplementary UV-B light for two hours each day for seven days had a greater effect on plant height, leaf area, fresh weight, and dry weight of sweet basil than one hour supplemental UV-B light [120]. In peppermint (*Mentha piperita*), combining white and UV-A or UV-B light increased the leaf area and leaf area index. In contrast, Johnson et al. [121] reported that the leaf area of sweet basil (*Ocimum basilicum*) was reduced by UV-B light. Another study reported that the combination of UV-B and white light affected neither the number of leaves nor the leaf area of Japanese mint (*Menthe arvensis*) [122].

It has been established that red, blue, and UV light can improve EO yield in various MAPs compared to white light. For instance, the effect of various LED lights (red, blue, red + blue (70% + 30%), or white) on some species of mint (*Mentha piperita*, *Mentha spicata*, and *Mentha longifolia*) was investigated by Sabzalian et al. [109]. The results showed that the EO content of all mint species was higher under red or combined red and blue light than under white light. In sweet basil (*Ocimum basilicum* L.), the EO content under blue light was 1.2–4.4 times higher than in plants cultivated under white light, and was lowest under red light, in experiments conducted at 50 μmol. m^−2^·s^−1^ PPFD for 70 days [112]. Changes in the EO composition of basil depended on light treatments. The second and third major compounds, respectively, were myrcene and linalool under blue light, and α-pinene and β-pinene under green and red light; under white light, these produced an intermediate response [112]. Even though UV light has a generally negative effect on plant growth, it might improve EO yield in some MAPs. For instance, Karousou et al. [123] studied the effect of supplementary UV-B radiation on the EO yield of two distinct chemotypes of spearmint (*Mentha spicata*) growing under field conditions. A significant increase in EO production was observed in chemotype II, while in chemotype I, the increase was insignificant. An increase in EO yield in Japanese mint (*Mentha arvensis*) was observed under UV-B or combined UV-A and UV-B light with white light [122]. Under the same conditions, increases in *l*-menthol and limonene concentrations were observed. Similarly, the combination of UV-A or UV-B with white light increased the EO yield in peppermint (*Mentha piperita*) [113,124]. There is limited information on the enhancement of EOs in MAPs under light [109]. However, the accumulation of EOs in MAPs may be due to the effects of light on the metabolic pathways of MAPs leading to an increase in EO yield [109,112]. Therefore, further investigations are needed to study the mechanisms that affect EO accumulation in MAPs under light, including UV light.

### 2.5. Pollution

Soil pollution is becoming a major environmental problem, due to rapid urbanization and industrialization [125]. Trace elements (TEs) are the major pollutants found in polluted soils. Soil pollution by TEs may occur naturally (i.e., erosion, geochemical background, volcanic eruption, etc.), but it mainly originates from anthropogenic activities (i.e., industrial and agricultural activities, traffic, mining, etc.) [126]. The most commonly found TEs in contaminated soil include arsenic (As), chromium (Cr), cadmium (Cd), lead (Pb), copper (Cu), zinc (Zn), nickel (Ni), and mercury (Hg) [127]. TEs cause significant risks to human health and the environment due to their non-biodegradability, bioaccumulation, and extreme toxicity [128,129]. Concerns have been raised about the accumulation of TEs in the soil, and their ability to penetrate and become incorporated into plants. These accumulated TEs adversely affect plant growth and productivity, as well as possibly contaminating the human food chain [129]. TEs can adversely affect MAPs’ development [130,131,132] as well as the quantity and composition of their secondary metabolites such as EOs [130,133].

Several studies have indicated that the response of MAPs to TEs is highly variable. This is probably due to the different experimental conditions (plant species, concentration, duration of treatment, composition of growth medium, culture conditions, etc.). The effect of TEs on plant growth has been observed in various Lamiaceae species. For instance, the uptake of As by the *Ocimum* spp. (*Ocimum tenuiflorum*, *O. basilicum*, and *O. gratissimum*) significantly decreased plant growth (plant height and dry weight) [134]. However, Cd, Pb, and Cu did not affect the growth of sweet basil (*Ocimum. basilicum*) [135]. Moreover, another study has shown that Cd, Pb, and Zn reduced root and shoot dry biomass by 15 and 10%, respectively, in garden sage (*Salvia officinalis*) [130]. Likewise, Raveau et al. [136] showed that the plant height and dry weight of clary sage (*Salvia officinalis*) were decreased by polluted soil (Pb, Zn, and Cd) compared with unpolluted soil. Similarly, a decrease in the dry biomass of shoot and root in chamomile (*Matricaria recutita*) was observed when grown in soil contaminated with Cd and Pb [137]. Despite being members of the same genus, menthol mint (*Mentha arvensis*), peppermint (*M. piperita*), and bergamot mint (*M. citrata*) responded differently to soil contaminated by Cr and Pb. *M. arvensis*’s shoot and root yields were not significantly affected by the application of Cr and Pb in the growth medium, whereas the shoot and root yields of *M. citrata* decreased, and those of *M. piperita* increased [131]. In addition, Sá et al. [138] reported that the application of Pb in soil did not affect the growth of garden mint (*Mentha crispa*). Another study reported that the application of Ni up to 30 ppm increased the shoot yield of *Tagetes minuta*. However, a further increase in the level of Ni decreased the shoot yield of that species [132]. Plant height and the number of branches per plant increased at low concentrations of Ni, while these parameters decreased with increasing Ni concentrations [132].

It has been hypothesized that the reduction in plant growth on TE-polluted soils is due to poor nutrient uptake because TEs compete with other plant nutrients [139]. Arsenate (AsV) accumulates in plants via the phosphate transporter (PHT) as an analog of phosphate (Pi), resulting in an inhibition of phosphorus (P) uptake [140,141,142]. Cd can reduce the uptake and translocation of essential nutrients including calcium (Ca), Cu, iron (Fe), Zn, and manganese (Mn) [143]. Cd is a divalent cation and may compete with plant nutrients for the same transporters [144]. Cd uptake by plants is mediated through cation transport systems, which are generally involved in the uptake of essential nutrients such as zinc–iron permease (ZIP) (ZRT–IRT-like proteins), natural resistant associated macrophage protein (NRAMP) transporters, or Ca channels and transporters [145]. In leaves, Cd penetrates via the Ca channel, disrupting the plant–water relationship [145], causing stomatal closure and leading to inhibition of photosynthetic activity, which subsequently reduces plant growth [143]. Furthermore, excessive Pb concentrations in soil inhibit plants’ uptake of minerals including Ca, Fe, Mg, Mn, P, and Zn, resulting in reduced plant growth [146].

The effects of TEs on the EO yields of many MAPs have been studied, including basil spp. (*Ocimum* spp.), mint spp. (*Mentha* spp.), lemon balm (*Melissa officinalis*), and sage (*Salvia officinalis*) [131,136,147,148]. Previous studies carried out on *Ocimum* species reported an increase in EO yield with the application of As [134,149]. Similarly, an increase in EO yield following Cd and Pb treatments was observed in sweet basil (*Ocimum basilicum* L.) [148]. Furthermore, differences in EO yield appeared between plant species within the same genus. The application of Cr and Pb in the medium growth, for example, decreased EO yields in menthol mint (*Mentha. arvensis*) and bergamot mint (*M. citrata*) while increasing the EO yield in peppermint (*M. piperita*) [131]. In contrast, the EO yield of *M. arvensis* was not affected by the application of Pb and Cd in soil [150], while in another mint species, garden mint (*Mentha crispa*), the EO yield increased when the concentration of Pb increased [138]. Moreover, the application of Pb in the soil decreased the EO yield of lemon balm (*Melissa officinalis*) [147]. In contrast, an increase in the EO yield of sage (*Salvia officinalis*) was observed in the presence of Cd, Pb, and Zn [130,136]. Differences in EOs’ chemical compositions were also observed. For example, menthol content in *M. arvensis* and *M. piperita* was not affected in response to Cr and Pb treatments, but minor constituents such as *α*-pinene and *β*-pinene were reduced in *M. arvensis* [131]. The reason for the changes in the EO yield of plants after the application of TEs is not known, but it might be attributable to the effect of TEs on enzymatic activity and carbon metabolism [68].

Phytoremediation is a green approach that relies on plants, including MAPs, to clean pollutants from contaminated soil [133]. It has been demonstrated through many studies that MAPs can be used in the phytoremediation of TE-polluted soils. For instance, vetiver (*Vetiveria zizanioides*) and rose (*Pelargonium roseum*) can act as phytoextractants for Pb. A study conducted by Chen et al. [151] showed that the formation of Pb-EDTA complexes (i.e., a chelating agent) in vetiver increased the accumulation of Pb in the roots and its translocation from roots to shoots. In addition, rose geranium (*Pelargonium roseum*) possesses a hyperaccumulator phenotype due to the accumulation of a high concentration of Pb in shoots relative to roots (8644 and 5550 mg Pb kg^−1^ DW, respectively) [152]. Other MAPs, including basil (*Ocimum* spp.), lavender (*Lavandula angustifolia* L.), sage (*Saliva sclarea*), and rosemary (*Rosmarinus officinalis* L.), act as multi-hyperaccumulators or excluders of multiple TEs including Cd, Zn, and Ni [52,153,154,155,156,157]. On the other hand, several MAPs have been shown to act as phytostabilizers for TEs. For example, palmarosa (*Cymbopogon martini*) can accumulate TEs (Cr, Ni, Pb, and Cd) with less translocation of TEs from roots to its aerial parts [158]. Furthermore, rosemary (*Rosmarinus officinalis* L.), clary sage (*Saliva officinalis*), and vetiver (*Vetiveria zizanioides*) can be useful for the phytostabilization of TEs such as Cu, As, Cd, Zn, and Pb [136,159,160].

In plants, the uptake and translocation of TEs occur through specific transporters such as ZIP (ZRT–IRT-like proteins), cation diffusion facilitators, heavy metal transport ATPases, metal transporter proteins, natural resistant associated macrophage proteins, and ATP-binding cassette transporters, which are localized at the plasma membrane and on the vacuole membrane of cells. These transporters facilitate TE uptake into plants and participate in their sequestration into vacuoles or cell walls [127,161,162]. In fact, plants have an antioxidant defense system which is triggered as a consequence of the increased level of reactive oxygen species (ROS). Glutathione (GSH) shows a high affinity for toxic metals, and plays an important role in metal sequestration and tolerance in plants [163,164,165].

## 3. Arbuscular Mycorrhizal Fungi Help MAPs to Cope with Abiotic Stress

It is evident from the studies cited above that water stress, salinity, temperature, light, and pollution are among the major abiotic stresses that can affect plant growth and consequently reduce biomass production and EO yield in some MAP species. In addition, abiotic stresses induced changes in the composition of distilled EOs. Since plants cannot move, they have evolved numerous strategies to cope with abiotic stress, among which AMF are known to promote plant growth and to confer better tolerance to abiotic stresses in host plants [31], including MAPs [32,33]. The following sections of the current review focus on the mechanisms used by AMF to alleviate the deleterious effects of abiotic stress in MAPs (Figure 1).

### 3.1. AMF Improves MAPs Growth and EOs Yield

Many studies have shown that AMF inoculation improves plant growth and biomass in addition to EO yield under abiotic stresses; see Table 3. Under conditions of water stress, Indian coral tree (*Erythrina variegate*) inoculated with *Funneliformis mosseae* presented a significant increase in plant growth parameters (shoot and root length, stem diameter, and leaf area) compared to non-inoculated plants [166]. The increase in dry biomass was explained by an improvement in water uptake by AMF extraradical hyphae [166]. Furthermore, sesbania (*Sesbania sesban*) colonized by different AMF species (*F. mosseae*, *Rhizophagus irregularis*, and *Claroideoglomus etunicatum*) showed an increase in fresh weight and lengths of shoot and root compared to non-inoculated plants [43]. The enhancement of MAPs’ growth by AMF could be attributed to AMF improving nutrient acquisition, especially P nutrition [167].

The accumulation of EOs in MAPs is generally attributed to glandular trichomes, which are specialized external secretory structures that secrete secondary metabolites including EOs [168,169]. Hence, there is a strong correlation between the density of glandular trichomes and EO yield. Interestingly, several studies revealed an enhancement in EO yield in many MAPs colonized with AMF, and these improvements were attributed to increases in the number of glandular trichomes in mycorrhizal plants [170,171,172]. It has been shown that the inoculation of oregano (*Origanum vulgare* L.) with *Glomus viscosum* increased the glandular density on the upper leaf epidermis [170]. Furthermore, sweet basil (*Ocimum basilicum* L.) inoculated with a mixture of AMF species, viz., *F. mosseae*, *Gigaspora margarita*, and *Gigaspora rosea*, showed an increase in EO yield [171]. This increase was associated with an abundance of glandular trichomes on the basal and central leaf zones [171]. Moreover, it has been reported that the inoculation of rose geranium (*Pelargonium graveolens*) with *F. mosseae* and *R. irregularis* increased the EO yield under conditions of water stress [37,173]. Similarly, basil (*Ocimum gratissimum* L.) plants inoculated with *R. irregularis* showed an increase in plant height and EO yield when experiencing water stress [44]. Likewise, sweet basil (*Ocimum basilicum* L.) inoculated with *R. irregularis* showed an increase in plant biomass and EO yield with a high level of metals in the soil [174]. The AMF can protect plants against TE toxicity by binding the metals in their hyphae, which limits the translocation of TEs from roots to shoots [174,175]. In addition, AMF hyphae produce glomalin, a glycoprotein that plays a role in the immobilization of TEs by generating protein–metal complexes, resulting in a reduction of TE content in the soil [176].

However, in addition to the advantages described above, some studies have reported that AMF can promote the content of active compounds in MAPs. Such benefits may be related to changes in the gene expressions involved in the biosynthesis of these compounds [177]. For example, Lazzara et al. [178] showed that *Hypericum perforatum* inoculated with a mixture of nine different AMF species increased the concentration of bioactive secondary metabolites such as hypericin and pseudohypericin in flowers, compared with non-inoculated plants under conditions of low P availability [178]. This enhancement could be attributed to the involvement of methyl jasmonate or salicylic acid in the mycorrhizal symbiosis, which can positively affect hypericin content [178,179].

**Table 3 foods-11-02591-t003:** Effect of mycorrhizal inoculation on MAPs growth and EOs yield under abiotic stress.

Stress Type	Plant Species	AMF Species	Growth	EO Yield	Reference
Water stress	*Ephedra foliata*	*Claroideoglomus etunicatum*, *Funneliformis mosseae*, *Rhizophagus irregularis*	+	n.d.	[33]
	*Erythrina variegata*	*Funneliformis mosseae*	+	n.d.	[166]
	*Foeniculum vulgare*	*Funneliformis mosseae,* *Rhizophagus irregularis*	+	n.d.	[180]
	*Glycyrrhiza uralensis*	*Rhizophagus irregularis*	+	n.d.	[40]
	*Lavandula spica*	*Funneliformis mosseae*, *Rhizophagus intraradices*	+	n.d.	[32]
	*Matricaria chamomilla*	*Funneliformis mosseae*	+	n.d.	[38]
	*Ocimum gratissimum*	*Rhizophagus irregularis*	+	+	[44]
	*Pelargonium graveolens*	*Funneliformis mosseae,* *Rhizophagus irregularis*	+	+	[37,173]
	*Ricinus communis*	*Funneliformis mosseae,* *Rhizophagus intraradices*	+	n.d.	[181]
	*Tagetes erecta*	*Glomus constrictum*	+	n.d.	[182]
Salinity	*Acacia nilotica*	*Glomus fasciculatum*	+	n.d.	[183]
	*Chrysanthemum morifolium*	*Diversispora versiformis Funneliformis mosseae*	+	n.d.	[184]
	*Ephedra aphylla*	*Claroideoglomus etunicatum*, *Funneliformis mosseae*, *Rhizophagus irregularis*	+	n.d.	[185]
	*Ricinus communis*	*Funneliformis mosseae,* *Rhizophagus intraradices*	+	n.d.	[181]
	*Sesbania sesban*	*Claroideoglomus etunicatum,* *Funneliformis mosseae,* *Rhizophagus irregularis*	+	n.d.	[43]
	*Valeriana officinalis*	*Funneliformis mosseae*, *Rhizophagus irregularis*	+	n.d.	[41]
High temperature	*Cyclamen persicum*	*Glomus fasciculatum*	+	n.d.	[186]
Trace elements	*Ocimum basilicum*	*Rhizophagus intraradices*	+	+	[174]
	*Trigonella foenum-graecum*	*Acaulospora laevis*, *Gigaspora nigra* *Glomus monosporum*, *Glomus clarum*	+	+	[187]

‘+’ indicates increasing responses; ‘–’ indicates decreasing responses; ‘=’ indicates no response; ‘n.d.’ indicates not determined.

Similarly, an enhancement in the concentration of artemisinin was observed in leaves of *Artemisia annua* inoculated with *R. intraradices*, and such an increase was attributed to the induced expression of the artemisinin biosynthesis genes including amorpha-4; 11-diene synthase, cytochrome P450, double bond reductase 2, and aldehyde dehydrogenase 1 [188]. Recently, Li et al. [189] reported that the inoculation of *Paris polyphylla* var. *yunnanensis* with mixtures of AMF species induced increased content of polyphyllin, which was associated with the expression of *PpSE*. A study caried out by Duc et al. [190] showed that a mixture of AMF species (*F. mosseae*, *Septoglomus deserticola*, and *Acaulospora lacunose*) promoted the accumulation of polyphenol content in *Eclipta prostrata* under salt stress [190]. This change in the polyphenol profile could be explained by the changes in the expression of genes involved in the polyphenol biosynthesis pathway [190]. Moreover, the AMF *R. irregularis* promoted glycyrrhizin and liquiritin in *Glycyrrhiza uralensis* plants experiencing water stress, and this enhancement was associated with an increase in the expression of glycyrrhizin biosynthesis genes (squalene synthase, β-amyrin synthase), P450 monooxygenase, and cytochrome P450 monooxygenase 72A154) [40].

### 3.2. AMF Improves Mineral Nutrient Uptake

It has been shown that AMF can boost the uptake of relatively immobile mineral nutrients in soils, including P, Zn, and Cu. The extraradical hyphae of AMF can extend beyond the depletion zone of the rhizosphere to absorb and transfer nutrient elements to the host plant [167]. Several studies have shown the impact of AMF inoculation on nutrient uptake in MAPs. For instance, two AMF species (*F. mosseae* and *R. irregularis*) significantly increased the leaf P concentration in fennel (*Foeniculum vulgare* Mill.) plants under different levels of water stress [180]. Under conditions of water stress, colonization of the roots by two AMF, *R. intraradices* and *F. mosseae*, alone or in combination, resulted in a significant increase in the concentrations of N, P, Fe, and Zn in rose geranium (*Pelargonium graveolens* L.) compared with non-inoculated plants [173]. It has been observed that water stress reduces P absorption, but this reduction rate was lower in mycorrhizal marigolds (*Tagetes erecta*) than in non-mycorrhizal plants [182]. Many studies have further shown that AMF inoculation enhanced the mineral nutrition of MAPs cultivated under abiotic stress. For example, it was shown that chamomile plants inoculated with *F. mosseae* demonstrated an increase in the shoot and root P and K concentrations under osmotic stress [38]. Similarly, enhancement of P, K, and Mg levels was reported in valerian (*Valeriana officinalis* L.) inoculated with *F. mosseae* and *R. irregularis* under salt stress [41]. In addition, hangbaiju (*Chrysanthemum morifolium*) inoculated with *F. mosseae* or *Diversispora versiformis* showed an increase in root concentration of N under salt stress [185]. Autochthonous *F. mosseae* (i.e., a drought-tolerant AMF strain) was found to be better than allochthonous *F. mosseae* (i.e., a drought-sensitive AMF strain) in terms of drought tolerance, with higher contents of N and K established in lavender (*Lavandula spica*) [32].

The main mechanism behind the higher nutrient concentration of mycorrhizal plants, especially in terms of P nutrients, is the increase in surface area for P uptake. The uptake of phosphate (Pi) from soil and AMF is mediated by the PHT1 gene family [191]. AMF induced Pi transporter genes during plant–AMF symbiosis [192]. AMF up-regulated the expression of phosphate transporter (PT) genes (*LePT4* and *LePT5*) in tomato plants to enhance plant tolerance under conditions of water stress [193]. Furthermore, AMF induced the expression of ammonium transporter protein and potassium (K^+^) transporter genes under conditions of water stress, which is crucial for N and K uptake by host plants in arid regions [194]. The secretion of acid phosphates by extraradical hyphae has also been proposed [195].

### 3.3. AMF Improves Plant Water Status

It has been demonstrated that AMF can significantly improve plants’ water uptake to alleviate abiotic stresses such as water stress, salinity, and light stress [196,197]. This increase in water uptake could be attributed to an increase in the hydraulic conductivity of the roots, which results from a change in root morphology [198,199]. The formation of external hyphae by AMF can also improve access to larger areas in soil, resulting in increased uptake and transport of water from soil to roots [200,201]. Mycorrhizal plants have improved access to soil water compared with non-mycorrhizal plants because the extraradical hyphae involved in water transport have a diameter of 2–5 µm, allowing them to easily penetrate small soil pores that are not accessible to root hairs [200,202]. It is noteworthy that there has been no research using MAPs to confirm water uptake by extraradical hyphae under abiotic stress.

Due to negative water potential, meaning low water availability in the soil, plants can be faced with the problem of acquiring a sufficient amount of water [203], a process that necessitates the involvement of aquaporins [204,205,206]. Aquaporins, also called water channel proteins, belong to the family of membrane channel proteins that facilitate the transport of water following an osmotic gradient [205,206,207]. Numerous studies on various plant species have indicated that AMF can modify the expression of genes’ coding for plant aquaporins under different environmental stresses [203,208,209]. Under salt stress conditions, common bean (*Phaseolus vulgaris*) roots inoculated with *R. irregularis* showed an up-regulation of three plasma membrane intrinsic protein (*PIP*) genes, *PvPIP1;1*, *PvPIP1;3*, and *PvPIP2;1*, compared with non-inoculated plants [208]. Under conditions of water deficit, soybean (*Glycine max*) and lettuce (*Lactuca sativa)* plants inoculated with *F. mosseae* and *R. irregularis* showed rapid decreases in the expression of certain *PIP* genes in comparison with non-inoculated plants [210]. A study carried out on liquorice (*Glycyrrhiza uralensis*), which is an important medicinal plant, found that inoculating plants with *R. irregularis* increased the expression of aquaporin genes when compared with non-inoculated plants, implying that AMF has a direct role in improving plant water status during water stress [40]. These studies clearly showed that the expression of aquaporin genes in host plants may respond differently according to leels of AMF colonization and stress imposed. Taken together, the up-regulation of aquaporin genes might be linked to an increase in the water transport capacity of plants [211], especially under abiotic stresses [208,209,210]. Further studies are required to analyze the expression of plant aquaporins in mycorrhizal MAPs exposed to different abiotic stresses, and to investigate the roles of aquaporins in facilitating plants’ water uptake.

Some studies have reported that mycorrhizal MAPs such as liquorice (*Glycyrrhiza uralensis*) [40], pangola-grass *Digitaria eriantha* [39], and *Polygonum cuspidatum* [212] showed an increase in stomatal conductance, thereby increasing their photosynthetic rate. Indeed, it has been well demonstrated that water stress prompts stomatal closure, causing a reduction in CO_2_ availability, which reduces photosynthesis in plants [100]. In this way, AMF could enhance photosynthesis by improving the plant’s water status, because the increase of stomatal conductance could result in an increase in gas exchange [213].

Under water stress, rose geranium (*Pelargonium graveolens* L.) inoculated with AMF species (*R. irregularis* or *F. mosseae*) showed higher water use efficiency (WUE) than non-inoculated plants [173]. Similarly, *R. irregularis* had a positive effect on WUE in liquorice (*Glycyrrhiza uralensis*) subjected to water stress [40]. The mechanisms behind the enhancement of WUE can be attributed to higher water uptake by extraradical hyphae [196,214], an increase of stomatal conductance and transpiration rate [196,215], and adjustment of osmotic potential [216].

### 3.4. AMF Modifies Endogenous Hormones

Abscisic acid (ABA) is a phytohormone that regulates plant development, including seed dormancy, inhibition of seed germination, growth regulation, and stomatal closure [217,218,219]. Its role in stress tolerance has received significant attention [220]. Indeed, ABA is considered a stress hormone due to its fundamental role in the responses of plants to abiotic stresses [221]. Under abiotic stress conditions, ABA levels increase, maintaining plant water status, regulating stomatal closure, and inducing changes in the expression of stress-inducible genes [217,221,222,223,224].

Several studies have documented that ABA levels were higher in mycorrhizal MAPs having experienced water stress [33,225]. For example, *Ephedra foliate* inoculated with *Claroideoglomus etunicatum*, *R. irregularis*, and *F. mosseae* showed an increase in ABA concentration under conditions of water stress compared with control plants [33]. Under dehydration, the leaves of mycorrhizal chile ancho pepper (*Capsicum annuum*) plantlets had higher ABA concentrations than non-mycorrhizal plantlets [225]. Thus, in stressed–inoculated plants, the modulation of ABA levels in guard cells induces the closure of the stomata in order to prevent water loss [226]. In contrast, AMF can also reduce the ABA levels in MAPs under conditions of water and salt stress [40,43]. For example, sesbania (*Sesbania sesban*) colonized by *F. mosseae*, *R. irregularis*, and *Claroideoglomus etunicatum* exhibited lower ABA levels than non-mycorrhizal plants under salt stress [43]. In addition, lower ABA concentrations were observed in the roots of liquorice (*Glycyrrhiza uralensis*) inoculated with *R. irregularis* compared with non-mycorrhizal plants under drought stress [40]. It has been reported that AMF could improve plants’ drought tolerance by decreasing root ABA concentration, which in turn up-regulated the expression of plasma membrane intrinsic proteins (*PIPs*) and antioxidant *SOD* genes [227]. Based on the above studies, it seems that the regulation of ABA levels depends on the AMF and the host plant species.

### 3.5. AMF Mediates Osmotic Adjustments

Osmotic adjustment is another important mechanism that allows plants to tolerate abiotic stresses including water stress, salinity, and osmotic stress. Osmotic adjustment allows the plant to maintain its turgor pressure and physiological activity by accumulating organic osmolytes including proline, soluble sugars, glycine betaine, and polyamines, as well as inorganic osmolytes such as K^+^, Ca^2+^, and Mg^2+^ [228,229]. Several studies have shown that AMF inoculation enhanced the stress tolerance of MAPs by increasing osmolyte accumulation [38,41,180]. Accumulation of soluble sugars in mycorrhizal plants leads to an adjustment of osmotic potential, constituting an important defense mechanism against abiotic stress [230]. Studies conducted on valerian (*Valeriana officinalis* L.) colonized with *R. irregularis* and *F. mosseae* indicated a higher accumulation of total soluble sugars in shoots and roots under salinity stress compared with non-mycorrhizal plants [41]. Compared with non-mycorrhizal plants, German chamomile (*Matricaria chamomilla* L.) plants inoculated with *F. mosseae* presented significantly higher levels of soluble sugars under osmotic stress [38]. An increase in soluble sugars was also reported in fennel (*Foeniculum vulgare* Mill.) plants inoculated with *F. mosseae* and *R. irregularis* during water stress [180]. Inoculation of *Ephedra foliate* with a mixture of AMF species, viz., *Claroideoglomus etunicatum*, *R. irregularis*, and *F. mosseae*, prompted a higher accumulation of glucose than in non-inoculated plants [33]. Recently, a study by Sun et al. [231] reported an increase in sucrose content in the roots of *Polygonum cuspidatum* inoculated with *F. mosseae*.

Sugar accumulation in mycorrhizal plants is due to an increase in photosynthesis activity and also due to the hydrolysis of starch [232]. Furthermore, an increase in the expression of sugar transporter encoding genes may also lead to sugar accumulation in mycorrhizal plants [213,233]. It has been reported that, during AMF symbiosis, the carbon allocation from source leaves to the roots increases, and that this process requires an increase in the expression of genes encoding for sugar transporters [234]. In the roots of tomato (*Solanum lycopersicon*) plants colonized by *F. mosseae*, a higher accumulation of sucrose and fructose content was observed. These results suggest that sucrose synthesized in source organs is loaded into the phloem. Therefore, the expression of the three genes encoding the sucrose transporters SUT1, SUT2, and SUT4 was increased [213]. However, as far as we know, the role of sugar transporter genes in the osmotic adjustment of mycorrhizal MAPs under abiotic stresses has not been reported, and further investigations are needed to elucidate this subject.

Proline is one of the most important osmolytes, and its high accumulation in response to abiotic stresses maintains the cell’s osmotic balance. Several studies reported an accumulation of proline in MAPs inoculated with AMF under abiotic stresses, such as chamomile (*Matricaria chamomilla*) [38], castor bean (*Ricinus communis*) [181], basil (*Ocimum gratissimum* L.) [44], *Ephedra alata* [184], *Ephedra foliate* [33], and valerian (*Valeriana officinalis* L.) [41]. However, some plants do not show this increase in osmolytes. For instance, the Indian coral tree (*Erythrina variegata* L.) inoculated with *F. mosseae* showed tolerance to water deficit by accumulating chlorophyll and carotenoids but not proline or soluble sugars [166].

### 3.6. AMF Stimulates Plants’ Antioxidant Defense Systems

Exposure of plants to abiotic stresses induces an accumulation of ROS including hydroxyl radicals (·OH), hydrogen peroxide (H_2_O_2_), singlet oxygen (O_1_^−^), and superoxide anion radical (O_2_^−^). An excess accumulation of ROS might be harmful to plants as a result of oxidative damage to proteins, lipids, and nucleic acids [235,236]. Malondialdehyde (MDA) is the end product of lipid peroxidation and is widely used as a marker of oxidative lipid damage [237,238]. The detoxification of excess cellular ROS requires an antioxidant machinery composed of enzymatic and non-enzymatic antioxidants. The enzymatic antioxidants comprise superoxide dismutase (SOD), peroxidases (POD), catalase (CAT), ascorbate peroxidase (APX), glutathione peroxidase (GPX), and glutathione reductase (GR), while the non-enzymatic system includes reduced GSH, ascorbic acid (AA), phenols, tocopherols, and carotenoids [235,236,239]. Several studies have shown that AMF colonization reduces oxidative damage by enhancing the antioxidant system in plants (including MAPs) under conditions of water stress, salinity, and heat stress. For instance, rose geranium (*Pelargonium graveolens* L.) inoculated with *F. mosseae* or *R. irregularis* presented lower accumulations of H_2_O_2_ and MDA in their leaves. These findings suggested that mycorrhizal plants had higher antioxidant enzyme activity (CAT, APX, GPX, and SOD) than non-mycorrhizal plants [37]. AMF-treated *Ephedra foliate* plants showed an increase in enzymatic (CAT, APX, GPX, SOD, and GR) and non-enzymatic antioxidants (GSH and AA), allowing them to maintain ROS levels that better prevent cell damage under conditions of water stress compared with non-mycorrhizal plants [240]. The activation by AMF of antioxidant activities enhanced tolerance to water stress. Autochthonous *R. irregularis* conferred greater drought tolerance to lavender (*Lavandula spica*) plants than allochthonous *F. mosseae* did, because higher antioxidant concentrations and greater development of intraradical and extraradical mycelium and arbuscular formation were produced in lavender inoculated with *R. irregularis* [32]. Sesbania (*Sesbania sesban* L.) [43] and *Ephedra aphylla* [184] colonized with a mixture of AMF species, viz., *Claroideoglomus etunicatum*, *R. irregularis*, and *F. mosseae*, showed an increase of different enzymatic antioxidants under salt stress. An enhancement in non-enzymatic antioxidants was also observed in sesbania (*Sesbania sesban* L.) [43]. A low level of MDA was observed in AMF-treated *Ephedra aphylla* plants compared with control plants [184]. Roots of cyclamen (*Cyclamen persicum* Mill.) plants colonized by *Glomus fasciculatum* subjected to heat stress presented an increase in the production of SOD and APX, as well as non-enzymatic antioxidants (AA), compared with non-mycorrhizal stressed plants [186].

AMF can also stimulate the production of non-enzymatic antioxidants other than GSH and AA, such as carotenoids, phenolic, and flavonoid compounds [37,182,184,241]. For example, rose geranium (*Pelargonium graveolens*) plants colonized by different *Glomus* species had higher phenol and flavonoid contents than non-colonized planted subjected to water stress [37]. Furthermore, *Ephedra aphyla* inoculated with a mixture of AMF, viz., *Claroideoglomus etunicatum*, *R. irregularis*, and *F. mosseae*, showed an increase in phenolic compounds under salt stress [241]. In addition, an enhancement in carotenoid content was documented in marigold (*Tagetes erecta*) plants colonized with *Glomus constrictum* [182]. These studies suggest that mycorrhizal MAPs can withstand abiotic stresses by maintaining ROS homeostasis through the accumulation of enzymatic and non-enzymatic antioxidants. Overall, the improvement in antioxidant defense systems caused by AMF could be attributed to the fact that AMF can accumulate ROS and also that AMF possesses *SOD* genes [242,243].

## 4. Synergistic Effects of AMF and Other Amendments on MAPs

### 4.1. Microbial Amendments

PGPR are soil bacteria living around or on the root surface, and in addition to AMF can also be used to improve MAPs’ growth under stress conditions. Indeed, PGPR are directly or indirectly involved in promoting plant growth and development via secretion into the rhizosphere soil of various regulatory chemicals (phytohormones, aminocyclopropane-1-carboxylic acid deaminase, and volatile growth stimulants) [244,245]. They also perform important biological functions, protecting plants against many pests or helping them to mitigate abiotic stresses such as drought, salt, or pollution [246,247].

Many studies have reported the beneficial effects of PGPR on the growth and yield of MAPs under drought stress. For example, Asghari et al. [248] showed that PGPR conferred drought tolerance and stimulated the biosynthesis of secondary metabolites in pennyroyal (*Mentha pulegium* L.) under water shortage conditions. The authors demonstrated high ABA, salicylic acid (SA), soluble sugars, phenolics, flavonoids, and oxygenated monoterpene contents, as well as radical scavenging activity in PGPR-inoculated plants under severe drought stress. The contribution of PGPR has also been shown to promote seed germination and root elongation of sage (*Salvia officinalis* L.) as well as growth of *Hyoscyamus niger* under water deficit conditions [51]. The improvement in seed germination rate could be due to an increase in gibberellin (GA) synthesis, which triggers the activity of α-amylase enzymes that promote early germination by improving starch assimilation [51,249]. This mechanism has been reported in hopbush (*Dodonaea viscosa* L.) under saline stress [250]. Likewise, plant growth promotion by PGPR involves the production of plant growth regulators such as indole acetic acid, GA, cytokinins, and ethylene [251,252,253,254]. In addition, through the production of biofilms, PGPR improve the availability of water for the plant, helping to manage abiotic stresses. Several works have reported modifications of MAPs’ metabolisms and increases in EO production using PGPR inoculants [255]. Indeed, PGPR are known to improve plant biomass and trichome production and to stimulate terpene biosynthesis, which improves EO production [49,51,256]. For example, increases of *cis*-thujone in sage (*Salvia officinalis* L.), α-terpineol in sweet basil (*Ocimum basilicum*) and marjoram (*Origanum majorana* L.), trans-sabinene in marjoram (*O. majorana* L.) and hybrid marjoram (*Origanum x majoricum*), and EO yield in fennel (*Foeniculum vulgare* Mill.) and geranium (*Pelargonium graveolens*) have variously been observed [49,51,257,258].

Further studies have demonstrated the beneficial and synergistic effects of dual PGPR and AMF inoculation on MAPs under metal, water, and salt-stressed conditions [259,260,261,262,263]. It has been demonstrated under conditions of water stress that the combined inoculation of myrtle (*Myrtus communis* L.) with AMF (*R. irregularis*) and PGPR (*Pseudomonas fluorescens, P. putida*) species improved hydromineral supply and EO production as well as enhancing plants’ oxidative defense [262]. Furthermore, it was reported that combined inoculation of PGPR and AMF improved plant growth, metabolite content, and root length under drought stress in various MAPs [264,265]. Similarly, co-inoculation with plant-growth-promoting fungi (like *Trichoderma* spp. and AMF) and PGPR has been reported to increase the production of secondary metabolites and hairy roots of Chinese salvia (*Salvia miltiorrhiza*) under conditions of drought or salinity stress [261].

### 4.2. Organic Amendments

Organic amendments including biochar, composts, manures, and mulch are known to limit biotic and abiotic stresses on plants by influencing the physico-chemical properties, nutrient availability, retention capacity, or microbial activity of the soil [48]. In addition, organic amendments are rich in humic acids, which decrease the availability of metals through adsorption and the formation of stable organometallic complexes [266], which could contribute to reducing TE stress for MAPs. If well-chosen, all these organic amendments could (depending on the MAP species and soil type) significantly improve soil fertility and nutrient cycling, nitrogen fixation, organic matter amount, plant growth, and consequently plant establishment under conditions of water and temperature stress [267,268]. The decomposition of organic amendments has also been reported to provide useful weed control for MAPs, and as a solution to avoid water runoff while maintaining soil moisture [269]. Thus, compost application increased the biomass and EO yields of black cumin (*Nigella sativa* L.), cumin (*Cuminum cyminum*), and sweet basil (*Ocimum basilicum* L.) under drought stress conditions, and of *Chenopodium album* L. in TE-polluted soil. These improvements in plant biomass could be due to an increase in NPK availability, the physical, chemical, and biological properties of soils, and increased soil cation exchange capacity and water retention, resulting in better plant growth [270,271]. The addition of biochar allows the immobilization of TEs such as Pb and Cd through various mechanisms including surface complexation, surface precipitation, electrostatic attraction, physical adsorption, and ion exchange, improving soil quality by increasing water retention and nutrient mineralization, and reducing soil bulk density [272]. Biochar-based amendments can improve plant biomass, stimulate production of pharmaceutically active compounds, including phenolic and flavonoids, consequently increasing antioxidant activities, and could reduce the risk of metal accumulation in the plant [272]. Indeed, biochar has been shown to improve the quality of bioactive compounds and the biomass yield of various MAPs, including kalmegh (*Andrographis paniculata*) cultivated under conditions of water stress, and brahmi (*Bacopa monnieri* L.), kalmegh, and ashwagandha (*Withania somnifera* L.) cultivated on TE-polluted soil [272,273].

Synergistic effects of combined organic amendment and AMF have been reported in many studies [274,275,276,277]. For example, the combination of AMF inoculation with organic amendment improved the biomass and active principle yields of lemon balm (*Melissa officinalis* L.) [274]. Similarly, Kaleji et al. [275] reported that the co-application of AMF and compost enhanced the growth and EO content of water mint (*Mentha aquatica*) [275].

### 4.3. Mineral Amendments

Mineral fertilizers such as NPK, silicate, or clay are known to affect soil porosity, pore distribution, water retention, and water availability in the soil [52,278,279]. Under conditions of water stress, the EO and biomass yields of sage (*Salvia officinalis* L.) have been reported to be significantly higher when fertilized with a conventional NPK mineral fertilizer [280]. Indeed, by changing soil pH, silicates reduced the phytoavailability of TEs, which are precipitated or chelated on soil aggregates [52]. Furthermore, silicate has been reported to be useful against various stresses, e.g., salinity, temperature, and flooding [281]. Generally, stress tolerance mechanisms can be attributed to physiological improvements in plants, activation of antioxidant systems, elicitation of secondary metabolites including ABA implicated in the osmotic stress response, or enzymatic activity (SOD, GPX, and CAT) [281]. For example, under stress conditions, silicate has been known to improve plant enzymatic activities such as SOD, CAT, GPX, and APX, as well as those of non-enzymatic antioxidants such as glutathione and proline, or plant transporters [281]. Likewise, clay amendment was reported to promote root elongation and drought tolerance in Pakchoi (*Brassica chinesis* L.) and hybrid sage varieties (*Salvia officinalis × S. pomifera, S. officinalis × S. tomentosa, S. officinalis × S. ringens* and *S. fruticosa × S. ringens*) under conditions of water stress [278,282], and silicate fertilizers reduced Cd transfer and accumulation in basil (*Ocimum basilicum*) shoots under metal stress [52].

Very few studies have demonstrated the synergistic effects of AMF inoculation and mineral amendments on MAPs under stress conditions. Indeed, AMF are often presented as an alternative to the application of chemical fertilizer in the production of MAPs. Recently, it has been shown that the application of a mixture of mycorrhizal species (*R. intraradices*, *F. mosseae*, *Glomus hoi*) and phosphorus fertilizer could improve the EO yield and physiological characteristics of peppermint (*Mentha piperita*) cultivated under different water stress conditions [283]. On the other hand, several studies have reported beneficial effects after the co-application of mineral and organic fertilizers [284,285]. For example, synergistic effects on fruit quality improvement have been described in the case of co-application of mineral and organic amendments under water deficit in pomegranate [286].

### 4.4. Biostimulants

Biostimulants are categorized as biological amendments and contain a wide range of molecules which may include chitosan, amino acids, humic substances, or plant extracts [46]. Their functional richness and potential interest in relation to MAPs grown under abiotic stress conditions have been highlighted in numerous studies [46,287,288,289,290]. It was shown that moderate water stress combined with SA application (300 ppm) increased EO yield [291] in lemon verbena (*Lippia citriodora* L.). The foliar application of SA and chitosan increased the EO content of thymbra (*Thymbra spicata* L.) and, particularly, the amount of thymol produced under reduced irrigation conditions [289]. Likewise, palm pollen grain extracts increased EO yield and the level of osmoprotectants (proline, amino acids, and soluble sugars) in basil (*Ocimum basilicum* L.) cultivated under drought conditions [290]. It has also been reported that the use of microalgae extracts caused a change in the composition of basil (*Ocimum basilicum* L.) and parsley (*Petroselinum crispum* L.) EOs, and increased the content of certain compounds [292]. Foliar spray of pluramin (a powdery compound of amino acids) was reported to reduce water stress damage in lemon balm (*Melissa officinalis* L.) [293].

To the best of our knowledge, no data have been reported on the synergistic effects of biostimulants and AMF inoculation on MAPs. On the other hand, some studies have demonstrated the beneficial and synergistic effects of combinations of various biostimulants, including those with organic amendments, on MAPs’ biomass production or EO yield [289,294]. For example, it was reported that co-application of biostimulants (such as *Azotobacter chroococcum* and *Azospirillum lipoferum)* and vermicompost improved the fresh and dry weights of coriander (*Coriandrum sativum* L.) [294].

## 5. Conclusions and Future Perspectives

With the increasing demand for natural products originating from MAPs, the cultivation of these plant species has become of great interest throughout the world. Unfortunately, MAPs face various environmental stresses due to anthropic activities, intensified by climate change, which have triggered harmful effects on their growth as well as the productivity and quality of their EOs. Therefore, much effort has gone into developing eco-friendly strategies that can tackle the problems related to ongoing climate change and improve plant tolerance. Hence, the actual challenge is to increase plant growth and biomass production while maintaining or improving the quality and yield of herbal materials such as EOs.

Microbial and organic or inorganic amendments can confer greater tolerance against abiotic stress in MAPs. AMF not only enhanced water and nutrient acquisition but also rapidly triggered the MAPs’ morphological, physiological, and molecular responses, which increased their ability to overcome the adverse effects of abiotic stress (Figure 1). It has been shown in the current review that AMF can improve plant growth as well as the quantity and quality of EOs. Many studies have demonstrated the synergetic effect of AMF with other soil amendments to improve MAPs’ growth and, consequently, EO yield and quality under abiotic stress. However, insights into the mechanisms behind the interaction of microbial soil amendments and MAPs under abiotic stress are very complex, and there remains a need for further investigation. Based on the data outlined in the current review, there is a need in the future for fundamental and applied investigations into different aspects:Most of the previously cited studies were conducted in the laboratory or under greenhouse conditions (i.e., pot experiments). However, further field experiments are required as many factors, including climate and microbial rhizosphere biodiversity, may influence the results compared with those obtained in controlled conditions.In studies where different strains were tested, the extent of AMF response on plant growth and root colonization varied with AMF species and also with the type and level of stress. Therefore, choosing the appropriate host plant and AMF species is important for using plant–AMF symbionts successfully. In future research, it will be important to screen indigenous and stress-tolerant AMF isolates to improve the effectiveness of arbuscular mycorrhizal symbiosis.More research should focus on the use of AMF in combination with PGPR or with organic or inorganic amendments to obtain more advantages in enhancing MAP growth and productivity. In addition, there is a considerable lack of data underlying the molecular mechanisms involved in the synergistic effects observed, and more generally in the modification of secondary metabolite pathway biosynthesis.Finally, various advanced techniques (proteomics, genomics, and metabolomics) could provide new insight into the mechanisms exerted by arbuscular mycorrhizal symbiosis, which confer stronger productivity and enhanced resistance to MAPs under abiotic stresses.

## Figures and Tables

**Figure 1 foods-11-02591-f001:**
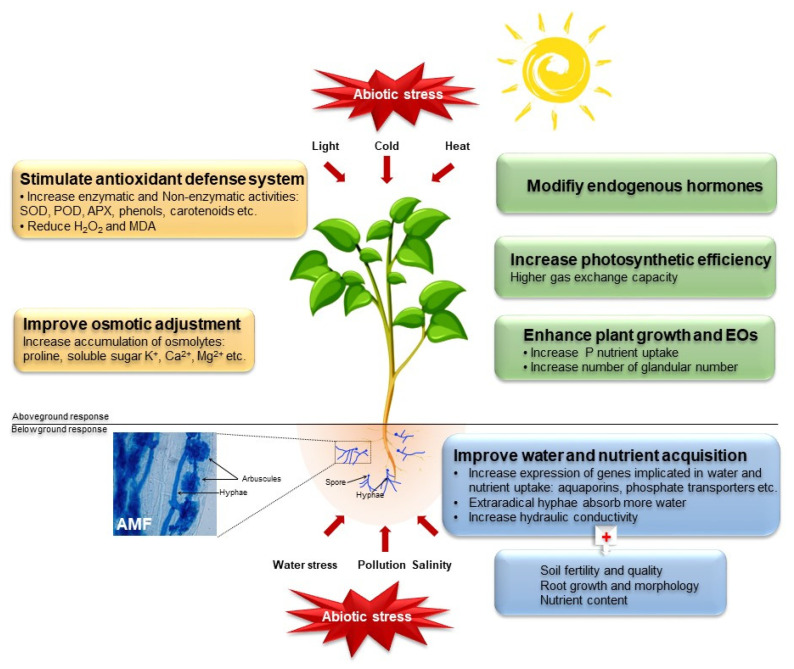
Mechanisms used by arbuscular mycorrhizal fungi (AMF) to improve stress tolerance and enhance the growth of medicinal and aromatic plants. AMF triggers the response of plants at morphological, physiological, and molecular levels to cope with the detrimental effects of abiotic stress. Arbuscular mycorrhizal symbiosis improves water and nutrient acquisition, enhances plant growth and abiotic stress tolerance. “+” indicates a positive effect. EOs, essential oils; SOD, superoxide dismutase; POD, peroxidase; APX, ascorbate peroxidase; H_2_O_2_, hydrogen peroxide; MDA, malondialdehyde; P, phosphorus.

**Table 1 foods-11-02591-t001:** Summary of water stress effects on growth and essential oil (EO) yield in various medicinal and aromatic plants (MAPs).

Family	Plant Species	Growth	EOs		Reference
			Yield	Plant Part Used for EOs Distillation	
Apiaceae	*Carum carvi*	–	+	Seeds	[56]
	*Coriandrum sativum*	–	n.d.	n.d.	[57]
	*Cuminum cyminum*	–	–	Seeds	[22]
	*Petroselinum crispum*	–	+	Roots; Leaves	[20]
	*Pimpinella anisum*	–	–	Seeds	[21]
Lamiaceae	*Lavandula angustifolia*	–	=	Leaves	[19]
	*Lavandula stoechas*	–	n.d.	n.d.	[53]
	*Mentha piperita*	–	–	Aerial parts	[58]
	*Ocimum basilicum*	–	+	Aerial parts	[59]
	*Ocimum americanum*				
	*Ocimum basilicum*	–	n.d.	n.d.	[54]
	*Rosmarinus officinalis*	–	–	Leaves	[60]
	*Salvia sclarea*	–	–	Arial parts	[61]
	*Saliva officinalis*	–	+	Arial parts	[62]
	*Salvia fruticosa*	–	+	Leaves	[19]
	*Sureja hortensis*	–	+	Aerial parts	[63]

‘+’ indicates increasing responses; ‘–’ indicates decreasing responses; ‘=’ indicates no response; ‘n.d.’ indicates not determined.

**Table 2 foods-11-02591-t002:** Summary of salt stress effects on growth and EO yield in various MAPs.

Family	Plant Species	Growth	EO		Reference
			Yield	Plant Part Used for EO Distillation	
Apiaceae	*Coriandrum sativum*	n.d.	–	Leaves	[74]
	*Coriandrum sativum*	–	+	Fruits	[75]
	*Coriandrum sativum*	–	+	Roots	[76]
	*Foeniculum vulgare*	–	+	Seeds	[72]
	*Trachyspermum ammi*	–	=	Seeds	[73]
Asteraceae	*Matricaria chamomila*	–	–	Flowers	[77]
Lamiaceae	*Calendula officinalis*	–	+	Flowers	[78]
	*Melissa officinalis*	–	–	Aerial parts	[79]
	*Mentha x piperita*	–	–	Aerial parts	[80]
	*Menhta piperita*	n.d.	–	Aerial parts	[81]
	*Mentha suaveolens*	–	–	Aerial parts	[80]
	*Ocimum basilicum*	–	+	Aerial parts	[81]
	*Origanum majorana*	–	–	Shoots	[82]
	*Saliva officinalis*	–	–	Leaves; Fruits	[83,84]
	*Satureja hortensis*	–	=	Aerial parts	[85]
	*Thymus daenensis*	–	n.d.	n.d.	[86]
	*Thymus maroccanus*	–	=	Aerial parts	[87]
	*Thymus vulgaris*	–	n.d.	n.d.	[86]
	*Thymus vulgaris*	–	+	Aerial parts	[88]
Poaceae	*Cymbopogon* *schoenanthus*	–	+	Aerial parts	[89]

‘+’ indicates increasing responses; ‘–’ indicates decreasing responses; ‘=’ indicates no response; ‘n.d.’ indicates not determined.

## Data Availability

Not applicable.

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
