# Peer review of "Significance of Arbuscular Mycorrhizal Fungi in Mitigating Abiotic Environmental Stress in Medicinal and Aromatic Plants: A Review"

_foods, 2022, doi:10.3390/foods11172591_

Round 1
Reviewer 1 Report
The present manuscript addresses a potentially interesting subject: are endomycorrhiza relevant to the development of medicinal and aromatic plants under stress conditions? In my opinion this implies the definition of what are Medicinal and aromatic plants, and in what respect they differ from the other plants under similar conditions. This is interesting because it would highlight the advantage of certain mechanisms (such as the production of essential oils) towards stress tolerance.
I would advice the authors to create this framework for a critical review of the subject.
In its present form the manuscript is a report that does not exploit the literature in an exhaustive manner and does not provide a rational to explain the different responses of medicinal and aromatic plants to stress under the presence of AMF.
Author Response
Reviewer #1:
The present manuscript addresses a potentially interesting subject: are endomycorrhiza relevant to the development of medicinal and aromatic plants under stress conditions? In my opinion this implies the definition of what are Medicinal and aromatic plants, and in what respect they differ from the other plants under similar conditions. This is interesting because it would highlight the advantage of certain mechanisms (such as the production of essential oils) towards stress tolerance. I would advise the authors to create this framework for a critical review of the subject.
Response: Thank you so much for your suggestions. We have mentioned it in the revised manuscript, in the introduction section.
Medicinal and aromatic plants (MAPs) are a diverse group of plant species with a great deal of interest due to their herbal medicine, pharmaceutical, cosmetic, and nutritional applications [1–3]. While medicinal plants have been referred to herbs used for therapeutic or pharmacological purposes, aromatic plants are rich in aromatic compounds and provide products that are widely used as spices, flavoring agents, and in cosmetics, and medicine[1,2]. These plant species are mainly cultivated for their various plant parts (roots, stems, leaves, flowers, seeds, buds, rhizomes, wood, bark, etc.) containing a wide range of secondary metabolites, including essential oils (EOs), which exert antimicrobial, antioxidant, and anti-inflammatory effects [4–6]. The cultivation of MAPs is looked upon not only as a source of reasonable health care products but also as a source of income and livelihood security [7].

Reviewer 2 Report
This review aims to provide a comprehensive view of how medicinal and aromatic plants respond to environmental stresses and how beneficial microorganisms, the arbuscular mycorrhizal fungi, can help medicinal and aromatic plants cope with these stresses. The authors provide a relatively comprehensive overview of relevant advances. However, I am aware that there have been numerous reviews on abiotic stresses in host plants by mycorrhizal fungi, such as the recent one by Cheng et al. (2021, Front. Plant Sci., 12:809473 doi: 10.3389/fmicb.2021.809473. You do not cite it), which has provided a good overview. I don't see any difference between your review and this literature, and the mechanisms obtained are almost equivalent. Maybe I think it is just a limited plant.
Similarly, Sun et al. (2021, Notulae Botanicae Horti Agrobotanici Cluj-Napoca, 49(3), 12454. https://doi.org/10.15835/nbha49312454) have just reviewed the interaction of arbuscular mycorrhizal fungi with medicinal plants. Sun et al. (2022, Front. Plant Sci. 13:818909. doi: 10.3389/fpls.2022.818909) revealed that mycorrhizae promoted the formation of medicinal plant components, which may be closely linked to the expression of related genes (Sun et al., 2022, Agronomy, 12, 1220. https://doi.org/10.3390/). Zhang et al. (2022, Front. Plant Sci., 13:840343. doi: 10.3389/fpls.2022.840343) also revealed the mycorrhizal properties of medicinal plants. Li et al. (2021, Phyton-International Journal of Experimental Botany, 90 (5): 1535-1547. doi:10.32604/phyton.2021.015697) also reported that the expression of PpSE was involved in mycorrhizal promotion of polyphyllin accumulation of Paris polyphylla) It should be noted that these works are somewhat informative to the current review.
Therefore, I would prefer to see the authors remove the medicinal plants and limit it to just the aromatic plants, or better illuminate the innovative aspects of the review. If the authors could not bear to remove it, I would prefer to see the authors analyze how mycorrhizae help medicinal and aromatic plants to promote the functional components of these plants under environmental stress, so that the review may be more meaningful than a single review of the physiological response of the plants. I think that medicinal and aromatic plants are more important for food functions rather than resistance.
The authors need to go deeper than just analyzing the mechanisms of association from a physiological perspective to dig into the possible molecular mechanisms, especially how mycorrhizae alter the functional constituents of these plants under stressful environments, which may be more desirable to readers.
The results of some early experiments covered by the authors that are not related to medicinal and aromatic plants can be appropriately removed and not heavily cited.
The authors are invited to focus on the above issues, especially the alteration of functional constituents of these plants under stress in response to mycorrhizal action and their potential mechanisms, although probably very little literature supports this.
Author Response
Reviewer #2:
- However, I am aware that there have been numerous reviews on abiotic stresses in host plants by mycorrhizal fungi, such as the recent one by Cheng et al. (2021, Front. Plant Sci., 12:809473 doi: 10.3389/fmicb.2021.809473. You do not cite it), which has provided a good overview.
Response: Thank you so much for your comment. We have mentioned and complete our manuscript with the missed reference in the introduction section, line 72, reference 46.
- I don't see any difference between your review and this literature (Cheng et al. (2021, Front. Plant Sci., 12:809473 doi: 10.3389/fmicb.2021.809473. You do not cite it), and the mechanisms obtained are almost equivalent. Maybe I think it is just a limited plant.
Response:
Thank you so much for your comment. Although it seems somewhat true that our review follows the previous review done by Cheng et al. (2021), specifically in the mechanism section. However, our review is still very different from the review of Cheng et al. (2021) in the numerous following aspects:
1) In the first section (impact of abiotic stresses on MAPs) we focus on the existing research literature (we searched a large number of publications) on the effect of abiotic stresses on MAPs, providing the reader an overall view of all studies carried out on MAPs. While Cheng et al. (2021) focused on the impact of drought stress on AMF diversity and activity but not on plants.
2) In the second section (Arbuscular Mycorrhizal Fungi Help MAPs to Cope with Abiotic Stress) we didn’t limit our review to the response of MAPs under drought stress (which is the case in Cheng’s review), but we extended the response of MAPs to other abiotic stresses such as salinity, low and high temperatures, light, and pollution.
3) Also, in our review, we focused only on the most important mechanisms of AMF-mediated abiotic stress tolerance in plants.
4) Furthermore, almost all of the studies used in the second section (Arbuscular Mycorrhizal Fungi Help MAPs to Cope with Abiotic Stress) are carried out on MAPs in order to provide the reader with a base on the topic that is sufficient to future research. These aspects are not found in Cheng‘s review, because after carefully reading this literature, we found that Cheng et al. (2021) discussed the effect of only one abiotic stress, drought, on AMF diversity and activity, and the mechanisms by which AMF enhance plant drought tolerance.
5) In addition, Cheng's review did not concern MAPs but other types of plants. There are a few examples of MAPs.
6) In our review, we added a subsection about the synergetic effect of AMF and other amendments on MAPs, which may provide the reader with an outlook for future research to enhance plant tolerance. This aspect has not been addressed by Chen et al (2021).
- Similarly, Sun et al. (2021, Notulae Botanicae Horti Agrobotanici Cluj-Napoca, 49(3), 12454. https://doi.org/10.15835/nbha49312454) have just reviewed the interaction of arbuscular mycorrhizal fungi with medicinal plants
Response:
Thank you so much for your comment. Now, in the revised manuscript, the cited reference Sun et al. (2021) has been added in the introduction section, line 61, reference 29. However, after reading carefully the suggested article, I do not feel that it matches our review that is still different. Indeed, in the suggested article, the authors focus on the physiological role of AMF in medicinal plants, and the mentioned mechanisms are not detailed more accurately. In addition, the authors discuss the diversity of AMF in the rhizosphere, as well as the interaction of AM-like fungus with medicinal plants. On the other hand, our review is different in several other interesting aspects, as mentioned in the previous response (please see comment 2).
- Sun et al. (2022, Front. Plant Sci. 13:818909. doi: 10.3389/fpls.2022.818909)
Sun et al., 2022, (Agronomy, 12, 1220. https://doi.org/10.3390/).
Zhang et al. (2022, Front. Plant Sci., 13:840343. doi: 10.3389/fpls.2022.840343
Li et al. (2021, Phyton-International Journal of Experimental Botany, 90 (5): 1535-1547. doi:10.32604/phyton.2021.015697). It should be noted that these works are somewhat informative to the current review
Response: Thank you so much for your suggestions. Now, in the revised version of the manuscript, all the cited references have been added.
- Sun et al. (2022, Front. Plant Sci. 13:818909. doi: 10.3389/fpls.2022.818909) revealed that mycorrhizae promoted the formation of medicinal plant components, which may be closely linked to the expression of related genes
Response: Thank you so much for your suggestion. We have added it (reference 235) in the revised manuscript. In the subsection “AMF Mediate Osmotic Adjustments”: line 601-603 (in track changes revised review)
- Sun et al., 2022, Agronomy, 12, 1220. https://doi.org/10.3390/).
Response: Thank you so much for your suggestion. We have added it (reference 215) in the revised manuscript. In the subsection “AMF Improve Plant Water Status”: line 544 (in track changes revised review)
- Zhang et al. (2022, Front. Plant Sci., 13:840343. doi: 10.3389/fpls.2022.840343) also revealed the mycorrhizal properties of medicinal plants.
Response: Thank you so much for your suggestion. We have added it (reference 35) in the revised manuscript. In the introduction section, line 65 in track changes revised review. However, the suggested article has been used to get new articles in order to update our review.
- Li et al. (2021, Phyton-International Journal of Experimental Botany, 90 (5): 1535-1547. doi:10.32604/phyton.2021.015697)
Response: Thank you so much for your suggestion. We have added (reference 192) it in the revised manuscript. In the subsection “AMF Improves MAPs Growth and EOs Yield “, line 440-442 in track changes revised review.
- The results of some early experiments covered by the authors that are not related to medicinal and aromatic plants can be appropriately removed and not heavily cited.
Response: Thank you so much for your suggestion. We included plants that are not related to medicinal and aromatic plants in our review because
- there is a lack of information related to MAPs
- and in some cases, if we do not use examples to support our idea, the discussion will be brief and superficial. In addition, we chose to provide a detailed discussion of the mechanisms used by AMF. This may provide a perspective for further research. However, you are right in that non-medicinal and aromatic plants should not be heavily cited. For that, we suppressed some examples from being prominent. (please see section “AMF Improve Plant Water Status”) Line: 519-526
- Therefore, I would prefer to see the authors remove the medicinal plants and limit it to just the aromatic plants, or better illuminate the innovative aspects of the review.
If the authors could not bear to remove it, I would prefer to see the authors analyze how mycorrhizae help medicinal and aromatic plants to promote the functional components of these plants under environmental stress, so that the review may be more meaningful than a single review of the physiological response of the plants. I think that medicinal and aromatic plants are more important for food functions rather than resistance. The authors are invited to focus on the above issues, especially the alteration of functional constituents of these plants under stress in response to mycorrhizal action and their potential mechanisms, although probably very little literature supports this.
Response: Thank you so much for your suggestion. Based on your suggestion, we have updated our literature review and added a few articles related to the alteration of active compounds by AMF in MAPs under environmental stresses. We also added the potential mechanisms at the molecular level (please see subsection “AMF Improve MAPs Growth and EOs Yield”. From Line 423 to 453

Reviewer 3 Report
The objective of this review was to report the effects of various abiotic stresses in medicinal and aromatic plants on growth, yield and composition of biologically active substances. Authors made an attempt to show some solutions on how to mitigate the effects of stresses. This is a comprehensive compendium of many results on medicinal plants especially aromatic in the context of the current many aspects of stress issues, including water stress, salt stress, low and high temperatures, light and pollution. The paper presents a review of the results of research showing the influence of mycorrhiza on herbs and the synthesis of oils and other substances. The literature on the topic is quite limited, so I admire the number of works found, which proves a very careful analysis and it also proves the growing importance of using mycorrhization also in the group of minor crops such as herbs. Subsection on synergistic effects of AMF and co-inoculations with bacteria and other fungi on herbs is very interesting source material for further research concerning mechanisms behind the interaction of microbial soil amendments under abiotic stress and there is a need for further investigation. This work include letter mistakes - the second part of the herb latin name should be written in lowercase, not in uppercase for example – line 249, 297, 299, 301, 305, 306, 308, 337. Line 357 please check the name Lavandula vera – is it Lavandula angustifolia ‘Vera’? line 748 – Thymbra – should be lowercase?
Authors should pay attention to correct text editing (bibliography) and English translation. After editing and naming corrections, the article may be published in the current version.
Author Response
Reviewer #3:
- This work includes letter mistakes - the second part of the herb Latin name should be written in lowercase, not in uppercase for example – line 249, 297, 299, 301, 305, 306, 308, 337. Line 357
Response: Thank you so much for your comment. Yes, there was mistake in the herb Latin name which is corrected in the revised review (in track changes revised review)
- Please check the name Lavandula vera – is it Lavandula angustifolia ‘Vera’? line 748 – Thymbra – should be lowercase?
Response: Thank you so much for your comment. We have addressed it during the revision of the manuscript. Lavandula vera: In Angelova et al. 2015, authors used the ancient name Lavandula vera, but we will take your comment into consideration. Thymbra is corrected to thymbra
- Authors should pay attention to correct text editing (bibliography) and English translation. After editing and naming corrections, the article may be published in the current version.
Response: Thank you so much for your comment. Now, in the revised version of the manuscript the text editing (bibliography) was corrected.

Reviewer 4 Report
This manuscript is a review on the effects of arbuscular mycorrhizal fungi in alleviating abiotic stresses in medicinal and aromatic plants.
The topic is relevant and it is of interest to the scientific community. The language is always understandable. The references are rather complete.
As such, only very few corrections are required:
L385, L418 and L465: replace “AMF Improves” with “”AMF Improve.
Figure 1: replace “Modifie endogenous hormones” with “Modify endogenous hormones”.
L449: replace “AMF triggers” with “AMF trigger”.
L520: replace “AMF Modifies” with “AMF Modify”.
L545: replace “AMF Mediates” with “AMF Mediate”.
L586: replace “AMF Stimulates” with “AMF Stimulate”.
L756: replace “To the best of our knowledge, there are no data reported the synergistic effects of biostimulants and AMF inoculation.” with “To the best of our knowledge, there are no data reported the synergistic effects of biostimulants and AMF inoculation on MAPs”.
L764 and L766: there is a repetition of the sentence: “…the cultivation of these plant species has become of great interest throughout the world”.
Author Response
Reviewer #4:
- As such, only very few corrections are required:
L385, L418 and L465: replace “AMF Improves” with “AMF” Improve.
L449: replace “AMF triggers” with “AMF trigger”.
L520: replace “AMF Modifies” with “AMF Modify”.
L545: replace “AMF Mediates” with “AMF Mediate”.
L586: replace “AMF Stimulates” with “AMF Stimulate”.
Response: Thank you so much for your comments. We have addressed it during the revision of the manuscript (in track changes revised review).
- Figure 1: replace “Modifie endogenous hormones” with “Modify endogenous hormones”.
Response: Thank you so much for your comment. We have replaced it with modify (Please see figure 1)
- L756: replace “To the best of our knowledge, there are no data reported the synergistic effects of biostimulants and AMF inoculation.” with “To the best of our knowledge, there are no data reported the synergistic effects of biostimulants and AMF inoculation on MAPs”.
Response: Thank you very much for your suggestion. We have replaced it in the revised manuscript (in track changes revised review). Line 798-800
- L764 and L766: there is a repetition of the sentence: “…the cultivation of these plant species has become of great interest throughout the world”.
Response: Sorry for the repetition. We have deleted it. Line 809-811 (in track changes revised review).

Round 2
Reviewer 2 Report
I think the authors have illuminated my concerns. Therefore, I would like to recommend the article for publication. Congratulations.